# Development of Next-Generation Probiotics by Investigating the Interrelationships between Gastrointestinal Microbiota and Diarrhea in Preruminant Holstein Calves

**DOI:** 10.3390/ani12060695

**Published:** 2022-03-10

**Authors:** Shih-Te Chuang, Chien-Ting Chen, Jui-Chun Hsieh, Kuan-Yi Li, Shang-Tse Ho, Ming-Ju Chen

**Affiliations:** 1Department of Veterinary Medicine, College of Veterinary Medicine, National Chung Hsing University, Taichung 402204, Taiwan; stchuang@dragon.nchu.edu.tw; 2Department of Animal Science and Technology, National Taiwan University, Taipei 106037, Taiwan; r08626006@ntu.edu.tw (C.-T.C.); jchsieh@ntu.edu.tw (J.-C.H.); doraemon102267@gmail.com (K.-Y.L.); 3Department of Wood Based Materials and Design, National Chiayi University, Chiayi 600355, Taiwan; stho@mail.ncyu.edu.tw

**Keywords:** next-generation probiotics, Holstein calves, microbiota, *Bifidobacterium longum* subsp. *longum*

## Abstract

**Simple Summary:**

The present study investigated the relationship between gastrointestinal microbiota and diarrhea in preruminant calves by using immune-related markers and further isolating specific bacterial strains, enriched in clinically healthy individuals, for potential next-generation probiotics. The gathering of microbiomic data strongly indicated the possible beneficial effects of *Bifidobacterium longum* subsp. *longum*. With further screening and isolating with immunomodulatory and antagonistic effects, two *Bifidobacterium longum* subsp. *longum* strains might be expected to emerge as next-generation probiotics. The finding here might provide a solution for preventing gastrointestinal disorders for preruminant calves without sustained periods of administration through inhibiting the infectious bacteria, immunomodulatory effect and possible modulating microbiota.

**Abstract:**

(1) Background: We aimed to isolate and identify potential next-generation probiotics (NGP) by investigating the interrelationships between gastrointestinal microbiota and diarrhea in preruminant Holstein calves. (2) Material and methods: Twenty preruminant Holstein calves were divided into healthy and diarrheic groups after the combination outcomes of veterinary diagnosis and fecal scores. The fecal microbiome, plasma cytokines, plasma immunoglobulin (Ig) G and haptoglobin were analyzed. The potential probiotic bacteria were identified by comparing the microbiota difference between healthy and diarrheic calves and correlation analysis with fecal scores and inflammatory markers. The identified bacteria were also isolated for further evaluation for antimicrobial activities and immunoregulatory effects. (3) Results: Microbiota analysis suggested that *Ruminococcaceae_UCG_014*, *Bifidobacterium* and *Pseudoflavonifractor* positively correlated with bovine IgG and negatively correlated with fecal score; inflammatory factors, bovine HP, and IL-8 were classified as beneficial bacteria contributing to the health of the calves. The alternation of gut microbial composition also induced changes in the functional gene enrichment of gut microbiota in calves. The gathering of microbiomic data strongly indicated the possible beneficial effects of *Bifidobacterium longum* subsp. *longum*, expected to develop as NGP. After isolation and evaluation of the potential functionality in vitro, two specific bifidobacterial strains demonstrated antimicrobial activities and immunoregulatory effects. (4) Conclusions: The results provide a new probiotic searching approach for preventing gastrointestinal disorders in preruminant calves. Further animal study is necessary to verify the results.

## 1. Introduction

Calf diarrhea, characterized by the frequent removal of soft feces, mostly impacts individuals under 6 weeks of age and their subsequent development [1]. The causes involve infectious agents, including viruses (bovine coronavirus, rotavirus, bovine viral diarrhea virus), bacteria (*Escherichia coli*, *Salmonella* spp.), and protozoa (*Eimeria zuernii*), which are mainly transmitted from the feces of infected animals to the mouths of susceptible animals [2]. The non-infectious parameters, such as stress, the nutritional and immunological conditions, and the production systems of young calves are also crucial for the incidence of diarrheic calves [2]. The treatment usually involves oral rehydration products and antibiotics. Accumulating evidence has indicated that the use of antibiotics in farm animals is associated with many adverse effects [3].

Several pieces of evidence suggested that the reconstitution of a healthy microbial community is an effective approach to prevent or treat gastrointestinal disorders [4]. The use of beneficial probiotics has been recognized to prevent the dysbiosis of intestinal microbiota and the establishment of pathogenic microbial populations. Many studies have highlighted that health and growth were the most positively pretentious responses to probiotic supplementation [5,6,7,8]. The most commonly used probiotic fed to young calves is live yeast, mainly *Saccharomyces cerevisiae* [9], and bacterial-based probiotics such as *Lactobacillus* spp., *Enterococcus* spp., and *Bacillus* spp. [10]. The feeding dosage varies depending on the probiotic strains and animal conditions, but sustained periods of administration are needed due to the fecal colonization of the probiotics decreasing with time.

However, a consensus regarding whether probiotics can in fact reduce the incidence of gastrointestinal diseases in young calves has not been reached. Alawneh et al. (2020) [11] indicated that, after a systematic review, there were no sufficient data to conclude that probiotic supplementation could provide a significant health benefit in the immunoregulatory effect and maintain gastrointestinal microbial balance. Most studies also specified that the exact mechanisms of the beneficial effects of probiotics on young calves still needed further investigation to precisely define and maximize the benefits that may be derived.

Kim et al. (2021) [4] demonstrated that intensive multi-donor fecal microbial transplantation (FMT) could change the gut microbiota of diarrheic calves with alterations in fecal microbial metabolite concentrations, suggesting that FMT may be a promising treatment for calf diarrhea beyond antibiotic-based therapies. However, since changes in the gastrointestinal microbiota are dynamic, defining a healthy calf’s microbiota is a difficult task. Probiotics isolated from preruminant Holstein calves according to the results obtained from next-generation sequencing (NGS) and bioinformatics analysis, which are the so-called next-generation probiotics (NGP), might provide a solution. Thus, in the present study, we aimed to investigate the interrelationships between gastrointestinal microbiota and diarrhea in preruminant Holstein calves, and identify and isolate the microbial biomarkers, associated with the healthy calves, as potential NGP. The antimicrobial activities and immunoregulatory effects of the isolated probiotic strains were also evaluated. Selecting probiotics originally from young calves, which might colonize the gastrointestinal track, could provide a solution for long-term health benefits without sustained periods of administration.

## 2. Materials and Methods

### 2.1. Animals and Sample Collection

All calves of the Holstein breed at a commercial dairy farm were housed in standard pens (1.5 m^2^) and administered maternal colostrum for the first three days after birth. From 4th day to 42nd day, the calves were equally fed with whole milk twice at 10% of the initial BW daily, 1/15th during 7th and 8th week via bottles. At day 14, all calves were gradually given solid starter feed (Calf Starter, Yi-Chen Ltd., Yi-Lai. Taiwan).

Ten healthy calves and 10 diarrheic calves between 30 and 51 days old (prior to weaning) were selected (Appendix A) by the outcomes of veterinary diagnosis and fecal scores. Blood and fecal samples were collected 2 h after the morning feeding for two consecutive days. Feces were collected per rectum from suckling calves by the veterinary team, and the samples were stored in a −80 °C freezer until analysis. Each sample was accurately weighed. Blood samples were collected from the jugular vein of the calves using a vacutainer tube. All animal experimental protocols were reviewed and approved by the Institutional Animal Care and Use Committee of the National Taiwan University (Approval No: NTU108-EL-00160).

### 2.2. Analyses of Fecal Score, Plasma Cytokines, Plasma IgG, and Haptoglobin

The fecal score was evaluated by the veterinary team using a calf health scoring guide written by the University of Wisconsin-Madison School of Veterinary Medicine [12]. Tumor necrosis factor (TNF)-α, interleukin (IL)-6 and IL-8 levels in the plasma were measured using commercial enzyme-linked immunosorbent assay kits (Bovine TNF-alpha, IL-6 and IL-8; DuoSet ELISA, R&D system, Minneapolis, MN, USA) according to the manufacturer’s instructions. For serum immunoglobulin (Ig) G and haptoglobin (HP) analysis, commercial enzyme-linked immunosorbent assay kits (Bovine IgG kit, R&D system Inc., Mckinley, MN, USA; Bovine HP, haptoglobin ELISA Kit, Fine test, Wuhan Fine Biotech Co., Ltd., Wuhan, China) were used.

### 2.3. Microbiota Analysis

The microbiota analysis adopted the process described by Huang et al. (2021) [13]. Briefly, the V3–V4 regions of the 16S rRNA gene were amplified with barcodes and sequenced by the Illumina MiSeq paired-end sequencing platform after extracting total genomic DNA. The Silva v.132 database, QIIME v1.7.0, and R v2.15.3 software [14] were used to analyze taxonomic annotation of each operational taxonomic unit (OTU), alpha diversity (Chao1 and Shannon), and beta diversity (principal component analysis, PCA) [15], respectively. The biomarkers were identified by linear discriminant analysis (LDA) effect size (LEfSe) algorithm. The phylogenetic investigation of communities was performed by reconstruction of unobserved states (PICRUSt v1.1.1) using the Kyoto Encyclopedia of Genes and Genomes (KEGG) database as reference genomes [16] to predict functional genes from the abundances of 16S rRNA sequencing data.

### 2.4. Isolation and Identification of Bifidobacterium longum subsp. longum

The strain isolation used the protocol described by Tanaka and Masahiko (1980) [17], with minor modifications. Briefly, 1 g fecal sample was homogenized. After 10-fold dilution with saline solution, 0.1 mL aliquots were cultured at 37 °C for 3 days anaerobically in modified Lactobacilli MRS agar (Acumedia, Neigen, Lansing, MI, USA) plates with 0.05% (*w*/*v*) L-cysteine hydrochloride monohydrate (Sigma-Aldrich Inc., St. Louis, MO, USA) and 0.01% cycloheximide (Sigma-Aldrich). The colonies with distinct morphologies were isolated. The Harrison’s disc method [18] was also used for colony selection. The isolates were then purified on modified MRS agar plates by streaking at least three times. Strains were kept at −80 °C in 15% (*w/w*) glycerol for further analysis.

Gram staining of each isolate adopted the procedure described by Colco (2006) [19]. After extraction of genomic DNA of each G+ strain, the full-length of 16S rRNA gene was amplified with the 8F and 15R primers [20] and sequenced with the 350F, 520R, and 930F primers [21]. Analyzing and assembling the sequences were conducted by Genomics BioSci & Tech Co., Ltd. (New Taipei, Taiwan) using Chromas version 2.23 (Technelysium Pty. Ltd., QLD, Australia), GENETYX v5.1 and GENETYX ATSQ v1.03 (Software Development Co., Tokyo, Japan). Phylogenetic trees were built by the neighbor-joining method [22]. Bootstrap analysis of 1000 replicates was applied to evaluate the statistical reliability of the trees using MEGA7 v7.0.14 software [23,24].

### 2.5. Antimicrobial Activity by an Agar Spot Test

Five pathogenic strains were used in this study. Bacillus cereus and Salmonella enterica were plated on Nutrient Broth (NB) agar (Fisher Scientific Ltd., Loughborough, Leicestershire, UK), while Staphylococcus aureus, Escherichia coli, and Vibrio parahaemolyticus were cultured using Tryptone Soy Broth (TSA; NEOGEN Corporation, Lansing, MI, USA), all at 37 °C with constant agitation. The antimicrobial activity was determined according to the process described by Tejero-Sariñena et al. (2012) [25].

### 2.6. Cytokine Production in RAW 264.7 Cells

The murine macrophage cell line (RAW 264.7, ATCC TIB71, American Type Culture Collection, Manassa, VA, USA) was cultured according to the procedure described by Hong et al. (2009) [26] with minor alterations. RAW 264.7 macrophages at a density of 5 × 10^5^ cells/mL were seeded in 1 mL medium and co-cultured with 10^6^ CFU/mL of each *Bifidobacterium* strain. A total of 50 ng of lipopolysaccharide (LPS; Merck-Sigma-Aldrich, Burlington, MA, United States) was added to 1 mL medium as the positive control. The levels of the cytokines (TNF-α and IL-10) in cell culture supernatants were measured by mouse cytokine ELISA kits (Invitrogen, Grand Island, NY, USA) after 24 h incubation at 37 °C.

### 2.7. Statistical Analysis

All phenotypic and next-generation sequencing (NGS) data were analyzed by a nonparametric Mann–Whitney U test to identify significant differences between groups. Bacterial networks and the correlation between bacterial biomarkers and diarrhea/inflammatory markers were constructed by Spearman’s correlation analysis. A *p*-value less than 0.05 is statistically significant. All statistical analyses were conducted by Statistical Analysis System v9.4 (SAS Institute Inc., Cary, NC, USA) and R software. 2.1 (R Core Team, Vienna, Austria).

## 3. Results

### 3.1. The Fecal Microbiota in Healthy and Diarrheic Preruminant Calves

Healthy and diarrheic calves (10 each) were selected according to the combinative results of clinical, systematic, and hematological examinations by the veterinarian and fecal score (>2). The diarrheic calves demonstrated significantly higher fecal scores (Figure 1A) than the healthy counterparts, as we expected. The significantly higher bovine HP and IL-8 and lower IgG observed in the diarrhea group, as compared with those of the healthy group (*p* < 0.05) (Figure 1B), indicate that the diarrheic calves were also under inflammatory status.

The results of the 16S rRNA analysis in the healthy calves and those with diarrhea show that a total of 871,579 and 871,107 effective tags were obtained from 1,042,854 and 1,041,882 raw paired-end reads, respectively. The alpha diversity, Chao1 richness estimator, and Shannon’s diversity index show no significant difference (*p* > 0.05) between the healthy calves and calves with diarrhea (Figure 2A). However, both groups could be clearly discriminated by the partial least squares discriminant analysis (PLS-DA) plot (Figure 2B). Thus, we further analyzed the dominant taxa at the family, genus, and species levels and found that the abundances of the family Bifidobacteriaceae and genera *Ruminococcaceae_UCG_014*, *Bifidobacterium* and *Pseudoflavonifractor* in the healthy calves were significantly higher than that of the counterparts with diarrheic conditions (*p* < 0.05) (Appendix A). Other genera, such as *Faecalibacterium* and *Prevoltella*_*9*, were also upregulated in the healthy group. Contrastingly, the abundances of the family Marinifilaceae and genera *Alloprevotella, Subdoligranulum, Ruminococcaceae*_*UCG*_*005*, *Ruminococcus*_*2*, *Lachnoclostridium*, *Odoribacter*, *Ruminiclostridium*_*9*, and *Lachnospiraceae*_*NK4A136*_group in the healthy calves were significantly lower than that of the diarrheic counterparts (*p* < 0.05). The abundance of the genus *Escherichia*_*Shigella* was also increased in the diarrheic calves. Regarding the top 20 species, the abundance of *Bifidobacterium longum* subsp. *longum* in the healthy group was significantly higher than that of the group with diarrhea (*p* < 0.05). While the level of *bacterium*_ *ic1277* in the healthy group was lower compared with those of the group with diarrhea (*p* < 0.05). The number of *Clostridium*_*sp*_*K4410MGS*_*306* increased in the diarrheic calves (*p* < 0.10) as well. We noticed that *Corynebacterium*_sp, *Bacteroidaceae*_*bacterium*_*DJF*_*B220* and *Corynebacterium*_*amycolatum* were only observed in the diarrheic calves with diarrhea.

### 3.2. Identification of the Critical Gastrointestinal Bacterial Biomarkers

Since the NGS results indicate that the health and diarrhea groups could be distinguished by gastrointestinal bacteria, the bacterial biomarkers were identified by the linear discriminant analysis (LDA) effect size (LEfSe) algorithm. A total of 19 influential taxonomic clades were recognized, including 1 order, 3 families, 13 genera and 2 species (Figure 2C). The most critical biomarkers identified in the healthy group were one order (Bifidobacteriales), one family (Bifidobacteriaceae), three genera (*Bifidobacterium*, *Ruminococcaceae*_*UCG*_*014* and *Pseudoflavonifractor*), and one species (*Bifidobacterium longum* subsp. *longum*). Contrastingly, 1 family (Marinifilaceae), 11 genera ([*Ruminococcus*]_*gauvreauii*_group, *Ruminiclostridium*_9, *Lachnoclostridium*, *Prevotellaceae*_*UCG*_*001*, *Odoribacter*, *Prevotellaceae*_*NK3B31*_group, *Ruminococcus*_2, *Lachnospiraceae*_*NK4A136*_group, *Subdoligranulum*, *Ruminococcaceae*_*UCG*_*005*, and *Alloprevotella*) and 1 species (*bacterium*_ *ic1277*) were the most influential taxa in the group with diarrhea. The results of the relative abundances of the bacterial biomarkers associated with the group with diarrhea are consistent with the above findings (Figure 2D). The relative abundances of all 12 bacterial biomarkers (genus and species levels) identified in the group with diarrhea were significantly higher than those of the healthy group (*p* < 0.05). Conversely, the relative abundance of all four bacterial biomarkers identified in the healthy group was significantly lower than that of the group with diarrhea (*p* < 0.05).

### 3.3. Correlation of Diarrhea and Inflammation Parameters with the Bacterial Biomarkers

The results of correlation (Figure 3A) show that the species enriched in the healthy group, such as *Bifidobacterium*, *Ruminococcaceae*_*UCG*_*014*, and *Pseudoflavonifractor*, were positively correlated with the bovine IgG, but negatively correlated with the fecal score and inflammatory factors, bovine HP, and IL-8. Conversely, the species enriched in the group with diarrhea, including [*Ruminococcus*]_*gauvreauii*_group, *Ruminiclostridium*_9, *Lachnoclostridium*, *Prevotellaceae*_*UCG*_*001*, *Odoribacter*, *Ruminococcus*_2, *Lachnospiraceae*_*NK4A136*_group, *Subdoligranulum*, *Ruminococcaceae*_*UCG*_*005*, and *Alloprevotella*, demonstrated negative correlations with the levels of IgG and positive correlations with the levels of fecal score, bovine HP, and IL-8.

In addition, we also clarified the co-occurrence patterns among bacterial biomarkers by constructing the bacterial network among 14 genera and two species, which was further correlated with the levels of IgG, IL-8 and fecal score. Among the genus bacterial network, [*Ruminococcus*]*_gauvreauii_*group, *Lachnoclostridium*, and *Prevotellaceae_UCG_001*, which negatively correlated with IgG (*p* < 0.05), were positively associated with five (Figure 3B), four (Figure 3C), and one (Figure 3D) other bacterial biomarkers in the group with diarrhea, respectively. In terms of the genus bacterial networks for fecal score and IL-8, *Ruminococcaceae*_*UCG*_*014* and *Pseudoflavonifractor*, the bacterial biomarkers identified in healthy calves, were positively correlated with each other and negatively correlated with five other diarrheic biomarkers (*p* < 0.05). Contrastingly, the bacterial biomarkers in the group with diarrhea, such as [*Ruminococcus*]_*gauvreauii*_group, *Ruminiclostridium*_9, *Odoribacter*, *Ruminococcus*_2, *Lachnospiraceae*_*NK4A136*_group, *Subdoligranulum*, *Ruminococcaceae*_*UCG*_*005*, and *Alloprevotella*, were positively and negatively associated with the certain diarrheic and healthy biomarkers, respectively.

### 3.4. The Relative Abundance of PICRUSt Functional Prediction of Fecal Microbiota in the Preruminant Calves

The results of the functional profiles indicate that the relative abundances of functional pathways in energy metabolism (proteasome, photosynthesis, photosynthesis proteins), metabolism of cofactors and vitamins (thiamine metabolism and cysteine), and amino acid metabolism (methionine metabolism) were significantly higher in the healthy group as compared with those of the diarrheic counterparts (Figure 4). Conversely, infectious diseases (*Staphylococcus aureus* infection), replication and repair (non-homologous end-joining), biosynthesis of other secondary metabolites (flavonoid biosynthesis), and xenobiotics biodegradation and metabolism (metabolism of xenobiotics by cytochrome P450, drug metabolism—cytochrome P450, chloroalkane and chloroalkene degradation, naphthalene degradation) and metabolism of terpenoids and polyketides (carotenoid biosynthesis) were significantly higher in the group with diarrhea compared with those of the healthy counterparts. These results suggest that the altered gut microflora not only influenced the healthy status in the preruminant calves, but it may also induce the increase in functional gene families in the gut of the calves.

### 3.5. Isolation and Identification of Bifidobacterium longum subsp. longum from the Healthy Preruminant Calves

First, 40 isolates were obtained from the feces of the preruminant calves using MRS medium and the Harrison disc method. After Gram staining and microscopic observation, nine Gram-positive isolates with similar phenotypic characteristics as compared with the type strain *Bifidobacterium longum* subsp. *longum* (BCRC14664) were selected for further identification (Figure 5A). The results of the comparative 16S rRNA gene analysis show that all nine LAB isolates belonged to the genus Bifidobacterium (Table 1). Of the nine, seven isolates (HCF-4, HCF-12, HCF-14, HCF-19, HCF-22, and HCF-24) shared 97% similarity with *Bifidobacterium longum* subsp. *Longum*, while the other two isolates (HCF-28 and HCF-30) had 97% similarity with *Bifidobacterium longum* subsp. *infantis*. Figure 5B shows the phylogenetic tree based on the 16S rRNA gene sequence analysis, depicting the phylogenetic relationships among the nine Bifidobacterium strains and seven strains obtained from the GenBank. *Escherichia coli* (U5 41) was used as the outgroup. The six strains, HCF-4, HCF-12, HCF-14, HCF-19, HCF-22 and HCF-24, were grouped together and formed a monophyletic clade, which showed a bootstrap value of 98% with *Bifidobacterium longum* subsp. *longum* (LC612559.1 and MT268981.1). Another strain, HCF-27, formed a monophyletic clade with *Bifidobacterium longum* subsp. *longum* (LC612559.1 and MT268981.1) with a bootstrap value of 98%. Two strains, HCF-28 and HCF-30, belonged to the *Bifidobacterium longum* subsp. *infantis*, pantothenic acid, and anacardic acid (*p* < 0.05) (Figure 5B), which are associated with purine metabolism and pantothenate and CoA biosynthesis (Table 1).

### 3.6. Determination of Antimicrobial and Immunomodulatory Effects

The results were compared against five pathogenic strains, as shown in Table 2. All seven strains of *Bifidobacterium longum* subsp. *longum* showed antagonistic effects against *Bacillus cereus*. The degrees of antagonism showed no difference among the strains (*p* > 0.05). Four strains, HCF-14, HCF-19, HCF-22, and HCF-24, inhibited the growth of *Salmonella enterica* without significant difference among the strains. For *Staphylococcus aureus* and *Vibrio parahaemolyticus*, six strains, except HCF-14, showed antimicrobial activity. However, none of the seven strains could inhibit the growth of *Escherichia coli*. The supernatants of each isolated strains were also evaluated. The results show that the supernatants of all seven strains had antimicrobial abilities against *Bacillus cereus* (Table 3). For *Salmonella enterica*, the supernatants of two strains, HCF-14 and HCF-19, also showed antagonistic effects. However, none of the supernatants from the seven strains had antimicrobial ability against *Staphylococcus aureus*, *Vibrio parahaemolyticus*, and *Escherichia coli*. Among the seven isolated strains of *Bifidobacterium longum* subsp. *longum* strains, HCF-19 was the most effective strain inhibiting the growth of the test pathogens. It showed inhibitory actions against four and two of the five test pathogens and their supernatants, respectively. In contrast, HCF-4 and HCF-12 were the least effective strains, showing antagonistic effects against two and one of the five test pathogens and their supernatants, respectively.

The results of the immunoregulatory effects show that the strain HCF-24 could significantly stimulate the production of proinflammatory cytokine, TNF-α, without LPS treatment, unlike the normal control (NC) group (*p* < 0.05). The upregulatory effect on TNF-α was not observed in the other six strains (Figure 6A). After LPS treatment, all seven isolated strains could significantly inhibit the levels of TNF-α compared with those of the LPS control (PC) (*p* < 0.05). For regulatory cytokines, IL-10, four strains, HCF-4, HCF-12, HCF-22, and HCF-27, could significantly stimulate the levels with or without LPS treatment compared with that of the LPS group (Figure 6B). The strain HCF-27 could induce in vitro proinflammatory cytokine, TNF-α, and inhibit its level after LPS treatment. This strain also induced regulatory cytokine, IL-10, with or without LPS treatment, indicating that this isolated strain might possess an immunoregulatory function.

## 4. Discussion

In the present study, we first investigated the interrelationships between gastrointestinal microbiota and diarrhea in preruminant Holstein calves. The physiological condition and diarrhea diagnosis of the calves were the key factors affecting the outcome. Thus, besides clinical and hematological examinations by the veterinarian, three immune-related markers, Hp, IL-8, and IgG, were also used to verify the diarrheic inflammatory status. Hp, one of the important acute phase proteins in ruminants [27], was related to the severity and activity of inflammation. Hp level in calves could be utilized for diagnosing bacterial [28] and viral [29] diseases. IL-8, known as an inflammatory marker, is a strong chemoattractant for T lymphocytes and polymorphonuclear leukocytes [13]. Bovine IgG has been reported not only binding to a wide range of pathogenic bacteria and viruses, but also to various allergens [30]. The non-sick calves feeding with IgG from 2 to 14 days of age decreased the frequency of incidence of diarrhea [31]. Thus, the concentrations of Hp, IgG, and IL-8 can be used as prediction indices of calf health [32] or prognose the course of the disease during rearing [33]. Our result was paralleled with previous findings, demonstrating that upregulation in the serum Hp concentration and IL-8 in the diarrheic calves were associated with clinical symptoms of diarrhea.

The finding in 16S rRNA indicates that the taxa with significant differences in abundances between the healthy and diarrheic calves were paralleled with LEfSe, which were identified as the critical gastrointestinal bacterial biomarkers. The most abundant and important commensal bacterial genus in the healthy calves was *Faecalibacterium*, which was also found in previous studies. The body weight gain during the pre-weaning period was reported to be significantly associated with the prevalence of *Faecalibacterium* spp. during the first week of a calf’s life [34]. The intervention with *Faecalibacterium prausnitzii*, a butyrate producer, reduced the incidence of severe diarrhea and related mortality rate in pre-weaned dairy heifers [35]. Our finding, accompanied with other previous studies, suggests a possible beneficial effect of *Faecalibacterium* spp. on the health and growth of young calves. *Bifidobacterium* has been reported to prevent gastrointestinal infections through competing for binding sites with pathogens and viruses on epithelial cells [36,37] and regulating the gastrointestinal immune system response [38]. Previous studies revealed that downregulation in the relative abundance of *Bifidobacterium* was involved in various dysbiosis-related intestinal diseases [39,40]. Our results also suggest a positive impact of *Bifidobacterium* on the prevention of gastrointestinal diseases in calves, with positive correlation with the bovine IgG and negative correlation with fecal score and inflammatory factors, bovine HP, and IL-8. *Pseudoflavonifractor*, positively correlated with the bovine IgG and negatively correlated with inflammatory factors in the present study, was reported to be beneficial to the immune homeostasis [41] and positively associated with the content of anti-inflammatory cytokines in yak calves [42]. The cell-free supernatant of *Pseudoflavonifractor* sp. AHG0008 showed the ability to suppress IL-8 secretion by peripheral blood mononuclear cells [43,44]. However, little was known about the role of *Pseudoflavonifractor* in the gastrointestinal sites.

Conversely, the diarrheic samples were categorized by a high abundance of sequences assigned to *Escherichia_Shigella*, the ileum mucosa-associated opportunistic pathogens. *Shigella* is phylogenetically distinct from several independent *E. coli* strains with four subgroups differentiated, which all cause shigellosis [45]. Thus, further identification of the ileum mucosa-associated *Escherichia_Shigella*, at species or strain level, is necessary. Other genera in the family of Rumminococcaceae (*Rumminococcaceae UCG 005*, *Ruminococcus*_*2* and *Ruminococcus*_*9*) were also enriched in the diarrheic calves. *Ruminococcaceae_UCG_005* was the most abundant genus at the post-weaning period [46] and at lactating Holstein dairy cows [47]. *Ruminococcus_2* has been reported to significantly reduce in probiotic-treated calves [46] and is potentially related to intestinal permeability and hepatic inflammation in a rodent model [48]. However, no evidence indicated the detrimental effects of these genera in calves. This result supports the high functional diversity reported within the Ruminococcaceae family and the difficulty to infer functions based on amplicon sequencing data [49].

The changes in gut microbial composition induced the functional gene family enrichment of gut microbiota in calves. More genes responsible for the energy and nutrient metabolisms were downregulated in the diarrheic calves with upregulating infectious diseases (*Staphylococcus aureus* infection), which was consistent with previous studies in calves [4] and humans [40]. Kim et al. [4] indicated that the improper metabolism of amino acids was one of the most important factors associated with diarrhea, suggesting that the interfering metabolisms were strongly connected to dysbiosis-related disease.

Moreover, broad upregulation in the poorly characterized two-component system and xenobiotics biodegradation and metabolism (metabolism of xenobiotics by cytochrome P450, drug metabolism—cytochrome P450) were observed in the present study. Two-component systems such as QseBC were mainly allocated in *Escherichia*, which produces a barrier effect against enteropathogens [50]. Other two-component regulatory systems, KdpDE, LiaS-LiaR, and GlnK-GlnL, are involved in the use of glutamine to sustain the homeostasis of stress response and cellular energy [51]. Poor characterization in two-component systems, suggesting a depleted beneficial metabolism from *Escherichia* in diarrheic calves, might play a crucial role in the development of diarrhea. The importance of the cytochrome P450 in the metabolism of endogenous compounds and exogenous compounds (drugs, natural products) has been well documented [52]. The enhancement of xenobiotics biodegradation and metabolism of cytochrome P450 may be due to the therapeutic treatment of diarrheic calves. Nevertheless, PICRUSt could just predict metagenomic function. Other omics, such as metabolomic approaches, are suggested to be able to recognize real changes in the metabolic function of the microbiota of diarrheic calves.

During the analysis in the interrelationships between gastrointestinal microbiota and diarrhea in preruminant calves, the current findings regarding the possible beneficial effects of *Bifidobacterium longum* subsp. *longum* appear promising, which are expected to develop as NGP. However, we acknowledged that a correlation study was unable to illustrate any underlying relationships. Thus, hypothesis-driven experiments were performed to evaluate the potential functionality of *Bifidobacterium longum* subsp. *longum* in vitro. Infectious agents, such as *Salmonella* spp. and *Escherichia coli*, are the major causes for calf diarrhea [2]. The following prevention or treatments with antibiotics, leading to an imbalance in the intestinal microbial community, could activate immune responses, inflammation, and peristalsis in the host gut, which also causes diarrhea [4]. The specific bifidobacterial strains, HCF-19 and HCF-27, presented here, may provide a solution for diarrhea prevention through the inhibition of infectious bacteria, an immunomodulatory effect, and possible modulating microbiota. Several studies have reported that *Bifidobacterium longum* could inhibit the growth of *Salmonella enterica*, *E. coli*, *Bacillus cereus*, and *Staphylococcus aureus* [53,54] due to the production of organic acids and specific bacteriocins. Certain *Bifidobacterium* spp. have also been used as a supportive treatment for diarrhea in dairy calves [55,56], but few of them were *Bifidobacterium longum* subsp. *lognum* and directly isolated from calves. It is worth noticing that the probiotic effects of *Bifidobacterium longum* subsp. *longum* were strain specific. Ibrahim et al. (2003) [57] also indicated that the beneficial effects of *Bifidobacterium* seem strain specific, not species specific.

## 5. Conclusions

In the present study, significant alterations in microbiota structure between healthy and diarrheic calves with deviations in the predictive metagenomic function of the bacterial communities and strongly correlating with immune-related markers provided a novel insight regarding the interrelationships between gastrointestinal microbiota and diarrhea in preruminant calves. Based on the microbiota findings, further screening and isolation of *Bifidobacterium longum* subsp. *longum* strains with immunomodulatory and antagonistic effects was conducted to characterize the relationship with the possible amelioration of diarrheic diseases. Two *Bifidobacterium longum* subsp. *longum* strains might be expected to emerge as next-generation probiotics. To the best of our knowledge, this is the first study investigating the relationship between gastrointestinal microbiota and diarrhea in preruminant calves using immune-related markers and further isolating specific bacterial strains, enriched in clinically healthy individuals for potential next-generation probiotics. Our results also suggest the importance of *Bifidobacterium longum* subsp. *longum* in the overall health of calves.

## Figures and Tables

**Figure 1 animals-12-00695-f001:**
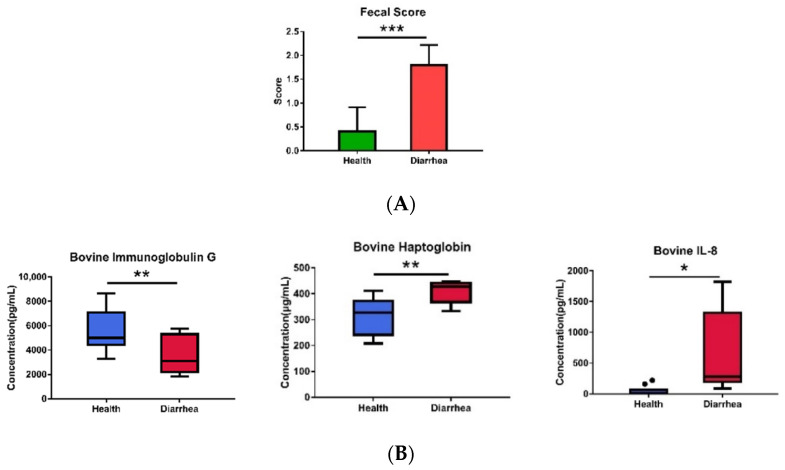
Evaluation of (**A**) the fecal score and (**B**) serum IgG, HP, and IL-8 in calves. The values are presented as the mean ± SD (*n* = 10) between groups. Data were analyzed by unpaired Student’s *t*-test. Significant difference: * *p* < 0.05, ** *p* < 0.01, *** *p* < 0.001. The dot represent the outlier value of samples.

**Figure 2 animals-12-00695-f002:**
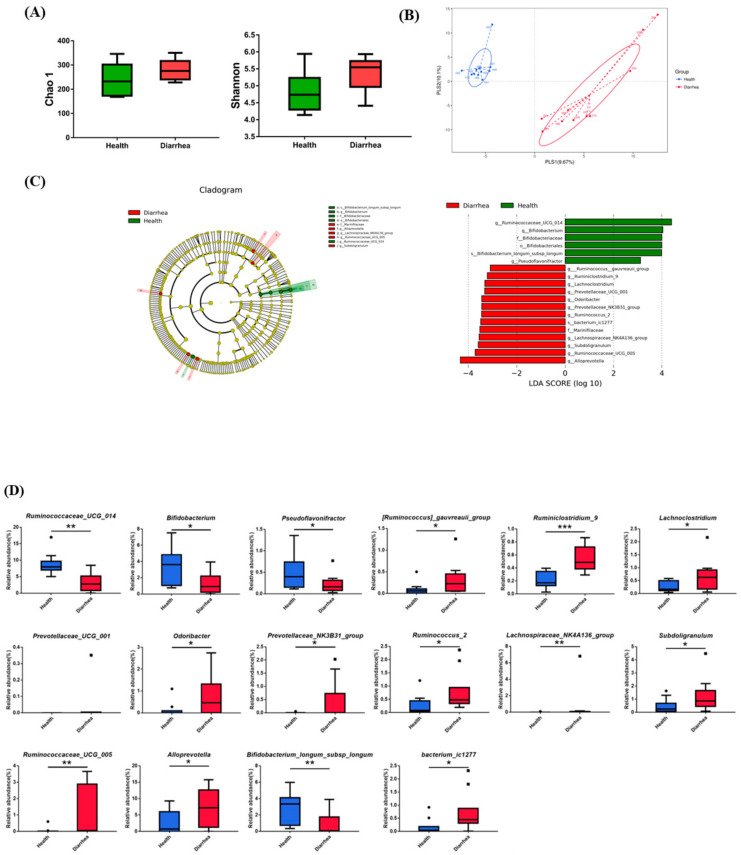
Microbiota analysis. (**A**) Bacterial community richness index estimated by the Chao1 value and community diversity index estimated by Shannon index between healthy and diarrheic calves. (**B**) Partial least squares discriminant analysis (PLS-DA) plots applied to maximally reflect the discrimination between healthy and diarrheic groups. (**C**) Taxonomic cladogram and histogram of the linear discriminant analysis (LDA) threshold value > 3.0 using LDA effect size analysis with the greatest differences in bacterial marker abundance between healthy and diarrheic calves. (**D**) Relative abundance of microbial markers at the genus and species levels of healthy (*n* = 10) and group with diarrhea (*n* = 10). Symbols indicate significant difference between two groups analyzed through Mann–Whitney U test. Significant difference: * *p* < 0.05, ** *p* < 0.01, *** *p* < 0.001.The dot represent the outlier value of samples.

**Figure 3 animals-12-00695-f003:**
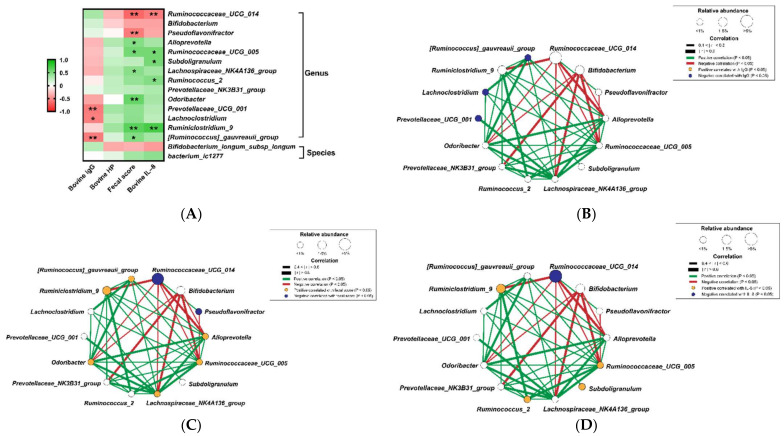
(**A**) Spearman’s correlation test between diarrheic biomarkers and calf gut microbial markers at genus and species level. Each cell was colored corresponding to Spearman’s correlation results. Significant difference: * *p* < 0.05; ** *p* < 0.01. Bacterial networks between calf gut microbial markers at genus level and their correlation with (**B**) bovine IgG, (**C**) fecal score or (**D**) IL-8 (Continued). Each node represents a genus biomarker, and the size of it corresponds to relative abundance. Colored nodes show significant correlation with IgG, fecal score or IL-8 (*p* < 0.05).

**Figure 4 animals-12-00695-f004:**
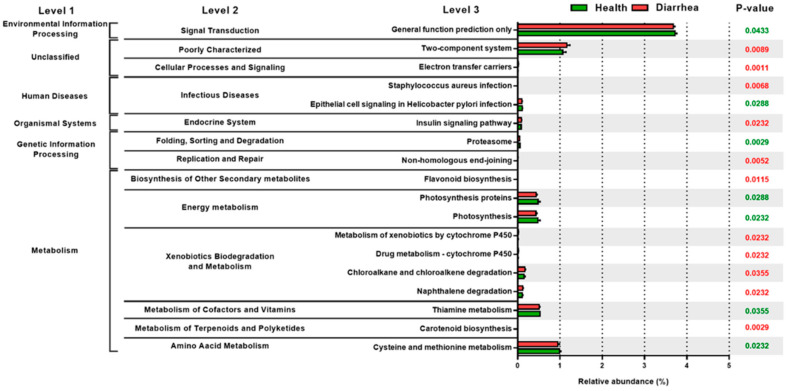
Comparison of the relative abundance of PICRUSt functional prediction of gut microbiota between healthy and diarrheic calves. The results represent means ± SD analyzed by Mann–Whitney U test. District gene categories are selected according to significant differences (*p* < 0.05) in KEGG pathway level 3.

**Figure 5 animals-12-00695-f005:**
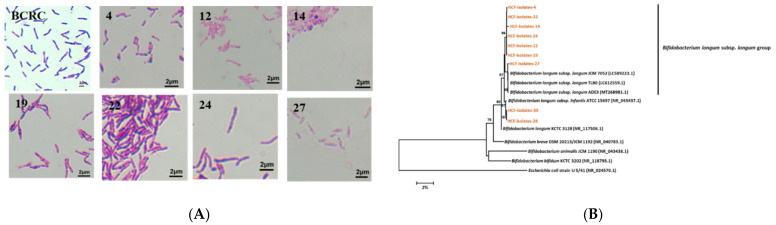
(**A**) Gram staining of *Bifidobacterium longum* subsp. *longum* from BCRC, HCF isolates-4, HCF isolates-12, HCF isolates-14, HCF isolates-19, HCF isolates-22, HCF isolates-24, and HCF isolates-27. (**B**) Phylogenetic tree based on the 16S rRNA sequence analysis constructed by the neighbor-joining method. *Escherichia coli* was used as the outgroup. The bootstrap values (expressed as percentage of 1000 replications) >70% are shown at branch points. Scale bar represents 2% sequence divergence.

**Figure 6 animals-12-00695-f006:**
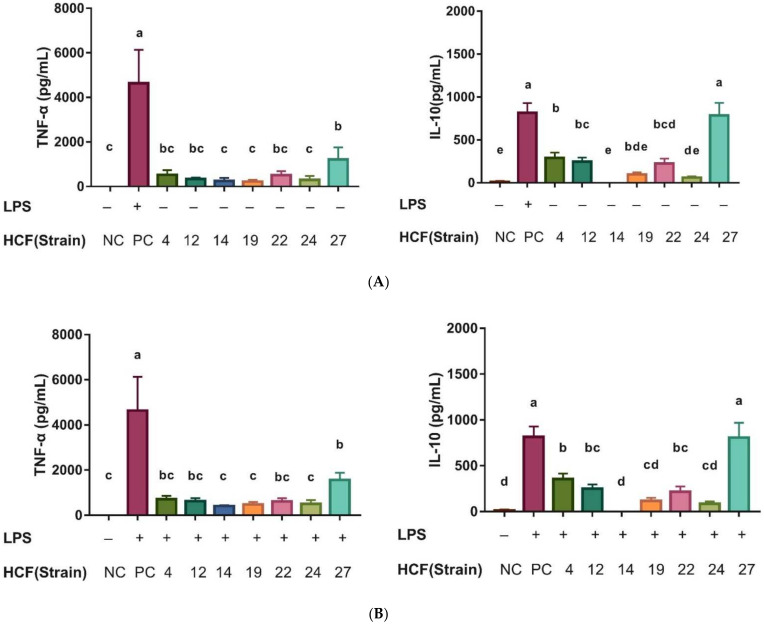
Effect of isolated bacteria strains in the secretion of (**A**) TNF-α and (**B**) IL-10 with or without LPS stimulation as co-cultured with RAW 264.7. Data were expressed with means ± SD (*n* = 3). + and − symbol represent with and without LPS addition, respectively. ^a,b,c^ Means for each group determined by a different superscript letter were significantly different analyzed by ANOVA with Tukey’s range test (*p* < 0.05). NC: negative control; PC: positive control; HCF (strain) 4-27: *B. longum* subsp. *longum* HCF4-27.

**Table 1 animals-12-00695-t001:** Identification of the nine lactic acid bacterial isolates using 16S rRNA gene sequences.

Isolate	The Nearest Matched Species from GenBank	Similarity (%)
HCF-4	*B. longum* subsp. *longum*	97
HCF-12	*B. longum* subsp. *longum*	97
HCF-14	*B. longum* subsp. *longum*	97
HCF-19	*B. longum* subsp. *longum*	97
HCF-22	*B. longum* subsp. *longum*	97
HCF-24	*B. longum* subsp. *longum*	97
HCF-27	*B. longum* subsp. *longum*	97
HCF-28	*B. longum* subsp. *infantis*	97
HCF-30	*B. longum* subsp. *infantis*	97

Similarity values were determined using the basic local alignment search tool (BLAST) of the GenBank. Sequences with ≥97% similarity to the previously published sequences were used as the criteria to indicate species identity.

**Table 2 animals-12-00695-t002:** The antimicrobial inhibition zone of (A) each isolated microorganism strain and (B) the antimicrobial activities against different pathogens.

(A)					
Inhibition zone of antimicrobial activity (mm)					
Isolated *B. longum* subsp. *longum* Strain	Pathogens				
	*Salmonella enterica*	*Bacillus cereus*	*Escherichia coli*	*Staphylococcus aureus*	*Vibrio parahaemolyticus*
HCF-4	0.00 ± 0.00 ^c^	9.83 ± 0.29 ^a^	0.00 ± 0.00 ^b^	12.33 ± 0.58 ^b^	10.00 ± 1.00 ^b^
HCF-12	0.00 ± 0.00 ^c^	10.00 ± 1.00 ^a^	0.00 ± 0.00 ^b^	12.83 ± 0.76 ^b^	10.83 ± 0.76 ^b^
HCF-14	10.37 ± 0.64 ^b^	10.67 ± 0.58 ^a^	0.00 ± 0.00 ^b^	0.00 ± 0.00 ^c^	0.00 ± 0.00 ^c^
HCF-19	10.73 ± 1.62 ^b^	10.33 ± 0.67 ^a^	0.00 ± 0.00 ^b^	11.67 ± 1.53 ^b^	10.07 ± 1.10 ^b^
HCF-22	11.83 ± 0.76 ^b^	9.37 ± 1.10 ^a^	0.00 ± 0.00 ^b^	12.50 ± 0.50 ^b^	10.53 ± 0.84 ^b^
HCF-24	11.50 ± 0.87 ^b^	10.73 ± 0.64 ^a^	0.00 ± 0.00 ^b^	12.00 ± 0.00 ^b^	9.33 ± 1.15 ^b^
HCF-27	0.00 ± 0.00 ^c^	10.33 ± 1.15 ^a^	0.00 ± 0.00 ^b^	13.00 ± 1.00 ^b^	10.00 ± 1.00 ^b^
PC (penicillin)	35.00 ± 1.70 ^a^	10.13 ± 1.22 ^a^	8.15 ± 1.24 ^a^	15.57 ± 1.52 ^a^	39.59 ± 2.88 ^a^
NC (broth)	0.00 ± 0.00 ^c^	0.00 ± 0.00 ^b^	0.00 ± 0.00 ^b^	0.00 ± 0.00 ^c^	0.00 ± 0.00 ^c^
(**B**)					
**Inhibition zone of antimicrobial activity (mm)**					
Isolated *B. longum* subsp. *longum* Strain	Pathogens				
	*Salmonella enterica*	*Bacillus cereus*	*Escherichia coli*	*Staphylococcus aureus*	*Vibrio parahaemolyticus*
HCF-4	–	+	–	++	+
HCF-12	–	+	–	++	++
HCF-14	++	++	–	–	–
HCF-19	++	++	–	++	++
HCF-22	++	+	–	++	++
HCF-24	++	++	–	++	+
HCF-27	–	++	–	++	++
PC (penicillin)	+++	++	+	++	+++
NC (broth)	–	–	–	–	–

Data were presented as mean ± SD. ^a,b,c^ Means for each group determined by a different superscript letter were significantly different analyzed by ANOVA with Tukey’s range test (*p* < 0.05). The symbol plus and minus means the ability of inhibition. “–“: diameter ≤ 8 mm, “+”: 8 mm < diameter ≤ 10 mm, “++”: 10 mm < diameter ≤ 20 mm, “+++”: >20 mm.

**Table 3 animals-12-00695-t003:** The antimicrobial inhibition zone of (A) the supernatant from each isolated microorganism strain and (B) antimicrobial activities against different pathogens.

(A)					
Inhibition zone of antimicrobial activity (mm)					
Supernatant of isolated *B. longum* subsp. *longum*	Pathogens				
	*Salmonella enterica*	*Bacillus cereus*	*Escherichia coli*	*Staphylococcus aureus*	*Vibrio parahaemolyticus*
HCF-4	0.00 ± 0.00 ^c^	10.50 ± 1.32 ^ab^	0.00 ± 0.00 ^b^	0.00 ± 0.00 ^b^	0.00 ± 0.00 ^b^
HCF-12	0.00 ± 0.00 ^c^	8.97 ± 0.06 ^ab^	0.00 ± 0.00 ^b^	0.00 ± 0.00 ^b^	0.00 ± 0.00 ^b^
HCF-14	11.63 ± 1.10 ^b^	10.67 ± 0.58 ^ab^	0.00 ± 0.00 ^b^	0.00 ± 0.00 ^b^	0.00 ± 0.00 ^b^
HCF-19	11.37 ± 0.55 ^b^	9.43 ± 1.01 ^ab^	0.00 ± 0.00 ^b^	0.00 ± 0.00 ^b^	0.00 ± 0.00 ^b^
HCF-22	0.00 ± 0.00 ^c^	10.83 ± 1.89 ^ab^	0.00 ± 0.00 ^b^	0.00 ± 0.00 ^b^	0.00 ± 0.00 ^b^
HCF-24	0.00 ± 0.00 ^c^	11.30 ± 1.13 ^a^	0.00 ± 0.00 ^b^	0.00 ± 0.00 ^b^	0.00 ± 0.00 ^b^
HCF-27	0.00 ± 0.00 ^c^	8.50 ± 0.50 ^b^	0.00 ± 0.00 ^b^	0.00 ± 0.00 ^b^	0.00 ± 0.00 ^b^
PC (penicillin)	35.00 ± 1.70 ^a^	10.13 ± 1.22 ^ab^	8.15 ± 1.24 ^a^	15.57 ± 1.52 ^a^	39.59 ± 2.88 ^a^
NC (broth)	0.00 ± 0.00 ^c^	0.00 ± 0.00 ^c^	0.00 ± 0.00 ^b^	0.00 ± 0.00 ^b^	0.00 ± 0.00 ^b^
(**B**)					
**Inhibition zone of antimicrobial activity (mm)**					
Supernatant of isolated *B. longum* subsp. *longum*	Pathogens				
	*Salmonella enterica*	*Bacillus cereus*	*Escherichia coli*	*Staphylococcus aureus*	*Vibrio parahaemolyticus*
HCF-4	–	++	–	–	–
HCF-12	–	+	–	–	–
HCF-14	++	++	–	–	–
HCF-19	++	+	–	–	–
HCF-22	–	++	–	–	–
HCF-24	–	++	–	–	–
HCF-27	–	+	–	–	–
PC (penicillin)	++	++	+	++	+++
NC (broth)	–	–	–	–	–

Data were presented as mean ± SD. ^a,b,c^ Means for each group determined by a different superscript letter were significantly different analyzed by ANOVA with Tukey’s range test (*p* < 0.05).The symbol plus and minus means the ability of inhibition. “–”: diameter ≤ 8 mm, “+”: 8 mm < diameter ≤ 10 mm, “++”: 10 mm < diameter ≤ 20 mm, “+++”: >20 mm.

## Data Availability

The datasets used and/or analyzed in the current study are available from the corresponding author on reasonable request.

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
