# Peer review of "Development of Next-Generation Probiotics by Investigating the Interrelationships between Gastrointestinal Microbiota and Diarrhea in Preruminant Holstein Calves"

_animals, 2022, doi:10.3390/ani12060695_

Round 1
Reviewer 1 Report
It is a very innovative, cutting-edge work. is carefull written and very interesting for the development of probiotics in preruminants.
I have no problem that once the grammar of the article is reviewed its posible its publication in the journal ANIMALS.
Tank you very much for taking me into account as a reviewer.
Author Response
Thank you.
Reviewer 2 Report
This is a nice piece of work, investigating microbiota composition in diarrheic claves and their association with immunoglobin and proinflammatory cytokines. Then they isolated probiotic bacteria and their immunomodulatory and antibacterial properties. The manuscript is well written. The study is useful. I have following suggestions.
- There are some grammars and English expressions issues, which sometimes limited the understanding of the manuscript. Authors need to edit the manuscript.
- Scientific name of the bacteria, protozoa, or any other organisms are required to italicize.
- L169: it is not clear what the cutoff values of the correlations were for the network analysis. A detailed information on the statistical analysis should be provided. I suggest to use high correlation coefficient values (P<0.01) for network analysis as some weaker correlations may not be confirmatory.
- L177-179: “The immune-related markers in serum, 177 including IgG, haptoglobin (HP) and proinflammatory chemokine (IL-8), were also deter- 178 mined.” Delete it. It is not the result.
- L182-185: “After verifying the physiological condition of the calves, we further investigated 182 whether the preruminant calves with diarrheic symptoms demonstrated different gastro- 183 intestinal bacterial components by comparing the fecal microbiota between the two 184 groups using NGS.” Delete. Not results, but methods. Please remove this type of sentences in the result section.
- L20: upregulation to abundances
- L363, 366, elsewhere: w/wo: write in full.
- Authors have mentioned ‘next generation probiotics’, but this has not been described anywhere. What is meant by next generation probiotics? It should be described to readers.
Well done work.
Author Response
This is a nice piece of work, investigating microbiota composition in diarrheic claves and their association with immunoglobin and proinflammatory cytokines. Then they isolated probiotic bacteria and their immunomodulatory and antibacterial properties. The manuscript is well written. The study is useful. I have following suggestions.
Response: Thank you.
Comments:
- There are some grammars and English expressions issues, which sometimes limited the understanding of the manuscript. Authors need to edit the manuscript.
Response: Thank you for the comment. We have checked the manuscript thoroughly and tried our best to revise all the mistakes.
- Scientific name of the bacteria, protozoa, or any other organisms are required to italicize.
Response: Thank you for the comment. We have checked the name of microorganisms (species and genus) carefully and modified the format.
- L169: it is not clear what the cutoff values of the correlations were for the network analysis. A detailed information on the statistical analysis should be provided. I suggest to use high correlation coefficient values (P<0.01) for network analysis as some weaker correlations may not be confirmatory.
Response: Thank you for the suggestion. For Spearman’s correlation test between diarrheic biomarkers and calf gut microbial markers at genus and species level, a p-value less than 0.05 (typically ≤ 0.05) is statistically significant. Both * P < 0.05 and ** P <0.01 are indicated in Figure 3A. We have rephrased the sentence to make it clear. (Please see the revised manuscript in lines 179-180)
- L177-179: “The immune-related markers in serum, 177 including IgG, haptoglobin (HP) and proinflammatory chemokine (IL-8), were also deter- 178 mined.” Delete it. It is not the result.
Response: This sentence was deleted. Thank you for the suggestion.
- L182-185: “After verifying the physiological condition of the calves, we further investigated 182 whether the preruminant calves with diarrheic symptoms demonstrated different gastro- 183 intestinal bacterial components by comparing the fecal microbiota between the two 184 groups using NGS.” Delete. Not results, but methods. Please remove this type of sentences in the result section.
Response: This sentence was deleted. We also removed this type of sentences in the result section.
- L200: upregulation to abundances
Response: The sentence has been modified as “The abundance of the genus Escherichia_Shigella was also increased in the diarrheic calves.” (Please see the revised manuscript in lines 210-211)
- L363, 366, elsewhere: w/wo: write in full.
Response: “w/wo” has been replaced by “with or without. (Please see the revised manuscript in lines 377, 380, 383)
- Authors have mentioned ‘next generation probiotics’, but this has not been described anywhere. What is meant by next generation probiotics? It should be described to readers.
Response: Thank you for the comment. The definition of NGP has been added in “Introduction”. (Please see the revised manuscript in lines 79-81)
“Probiotics isolated from preruminant Holstein calves according to the results obtained from next generation sequencing (NGS) and bioinformatics analysis, which is so called next generation probiotics (NGP), might provide a solution.”
Reviewer 3 Report
Dear Authors
I consider the manuscript presented me for review as interesting, innovative and it has application for practice. The manuscript has a clearly defined hypothesis and goal, and the conclusions are adequate to the obtained results. The only complaint is an editorial error - no line numbering.
Author Response
I consider the manuscript presented me for review as interesting, innovative and it has application for practice. The manuscript has a clearly defined hypothesis and goal, and the conclusions are adequate to the obtained results.
Response: Thank you.
Comments:
- The only complaint is an editorial error - no line numbering.
Response: Thank you for the comment. The line number has been shown in the revised manuscript.
Reviewer 4 Report
Dear authors,
The present manuscript is well presented.
Nevertheless eligible criteria of the participants and criteria for the allocation are presented adequately.
Is the sample size of 20 animals strong enough for robust conclusions?

Author Response
The present manuscript is well presented. Nevertheless eligible criteria of the participants and criteria for the allocation are presented adequately.
Response: Thank you.
Comments:
- Is the sample size of 20 animals strong enough for robust conclusions?
Response: Thank you for the comment. We have rephrased the conclusion as “Our results also suggested the importance of Bifidobacterium longum subsp. longum for the overall health of calves.” (Please see the revised manuscript in lines 502-503)